# Modeling pastoralist movement in response to environmental variables and conflict in Somaliland: Combining agent-based modeling and geospatial data

Erica L. Nelson[1,2]*, Saira A. Khan[1,3], Swapna Thorve[4], P. Gregg Greenough[1,2,3]

**1** Harvard Humanitarian Initiative, Harvard University, Cambridge, Massachusetts, United States of America, **2** Division of Global Emergency Medicine and Humanitarian Programs, Department of Emergency Medicine, Harvard Medical School, Brigham and Women's Hospital, Boston, Massachusetts, United States of America, **3** Harvard T.H. Chan School of Public Health, Harvard University, Boston, Massachusetts, United States of America, **4** Department of Computer Science, University of Virginia, Charlottesville, Virginia, United States of America

\* elnelson@partners.org

**Data Availability Statement:** All data utilized in this study are open source and details of accessing

## Abstract

Pastoralism is widely practiced in arid lands and is the primary means of livelihood for approximately 268 million people across Africa. Environmental, interpersonal, and transactional variables such as vegetation and water availability, conflict, ethnic tensions, and private/public land delineation influence the movements of these populations. The challenges of climate change and conflict are widely felt by nomadic pastoralists in Somalia, where resources are scarce, natural disasters are increasingly common, and protracted conflict has plagued communities for decades. Bereft of real-time data, researchers and programmatic personnel often turn to post hoc analysis to understand the interaction between climate, conflict, and migration, and design programs to address the needs of nomadic pastoralists. By designing an Agent-Based Model to simulate the movement of nomadic pastoralists based on typologically-diverse, historical data of environmental, interpersonal, and transactional variables in Somaliland and Puntland between 2008 and 2018, this study explores how pastoralists respond to changing environments. Through subsequent application of spatial analysis such as choropleth maps, kernel density mapping, and standard deviational ellipses, we characterize the resultant pastoralist population distribution in response to these variables. Outcomes demonstrate a large scale spatio-temporal trend of pastoralists migrating to the southeast of the study area with high density areas in the south of Nugaal, the northwest of Sool, and along the Ethiopian border. While minimal inter-seasonal variability is seen, multiple analyses support the consolidation of pastoralists to specifically favorable regions. Exploration of the large-scale population, climate, and conflict trends allows for cogent narratives and associative hypotheses regarding the pastoralist migration during the study period. While this model produces compelling associations between pastoralist movements and terrestrial and conflict variables, it relies heavily on assumptions and incomplete data that are not necessarily representative of realities on the

these data are in Supplemental information 2 of the manuscript.

**Funding:** This project was funded by the National Center for Civic Innovation Inc (https://www.fcny.org/fcny/about/ncci/) acting through and on behalf of The Governance Lab ("GovLab"; https://www.thegovlab.org/). The project grant (no available grant #) funded PGG, ELN, and SAK to conduct this research. This project was originally sponsored by the World Bank (https://www.worldbank.org/). The funders had no role in study design, data collection and analysis, decision to publish, or preparation of the manuscript.

**Competing interests:** Regarding our Competing Interests, the authors of this publication have received funding from the National Center for Civic Innovation Inc., Wellspring Philanthropic Fund, Brigham and Women's Hospital, South Shore Hospital, L'Ecole des hautes etudes en sante publique, and the Harvard School of Public Health. We have engaged in no consultancy, patents, or product production. This does not alter our adherence to PLOS ONE policies on sharing data and materials. It has not influenced the outcomes of this research.

ground. Given the paucity of data regarding pastoralist decision-making and migration, validation remains challenging.

## Background

Pastoralism is widely practiced in arid and semi-arid lands (ASALs) and is the primary means of livelihood for approximately 268 million people across Africa [1]. Pastoral mobility is largely driven by the availability and quality of fodder and water to maintain livestock herds, influencing the spatial and temporal variability of pastoral migration patterns across landscapes [2,3]. While the non-sedentary lifestyle of pastoralism can be considered an effective adaptation technique to environmental changes, dependence on natural resources contributes to the risk-averse nature of pastoralism [1,2]. Unpredictable climatic changes contribute to the increase in the severity and frequency of natural hazards, irregular rainfall patterns, extreme temperatures, and changing land cover, affecting the availability of natural resources required to support livestock herds [3–5]. These environmental effects are compounded by ongoing land degradation, privatization, conflict, and numerous other factors that weaken pastoral systems [1,5].

The challenges experienced by pastoralist communities are unevenly felt across the continent but are particularly pertinent in Somalia, where approximately 65% of the population relies on pastoralism as a primary livelihood [6]. In the last decade alone, Somalia has experienced numerous devastating environmental shocks over a short period of time. Between 2010 and 2012, a catastrophic drought and subsequent food insecurity and famine affected 13 million people across the Horn of Africa, many of whom were pastoralists [7]. Of all the countries affected by the drought, Somalia was arguably the hardest hit, a situation that was aggravated by the inability to provide timely assistance due to instability, conflict, and lack of humanitarian coordination. Shortly after, Somalia experienced another drought between 2016 and 2017 and lower-than-normal rainfall through the end of the decade, resulting in growing numbers of internally displaced populations (IDPs). The Food and Agriculture Organization (FAO) [1] noted that it takes approximately five years for a livestock-dependent household to fully recover from the effects of severe drought. The severe droughts in Somalia during 2010–2012 and 2016–2017 then theoretically did not give pastoral households time to fully recover. A variety of adaptation techniques are used to protect livestock-related livelihoods in times of drought or conflict, including forming agreements and alliances with members of the community, sharing of necessary resources, and diversification of livelihoods [8]. At times, however, the expansion of private land and environmental degradation results in resource conflicts and occasional livestock death.

The impact of environmental hazards is compounded by ongoing conflict and political instability that has troubled the country for decades. Ongoing armed conflict between the Somali militia, the African Union Mission in Somalia, and non-state actors has led to widespread insecurity and displacement, particularly in Southern Somalia [9]. In the northern part of the country, communal violence prompted by clan/ethnic differences and the quest for political autonomy has been an ongoing problem in the contested regions of Puntland and Somaliland. Somaliland seeks international recognition as an independent country while disputing territorial borders with neighboring Puntland [4]. In 2018 alone, an estimated 578,000 people were displaced due to conflict, while an additional 548,000 people were displaced by regional disasters in Somalia [9]. To date, the total number of IDPs in Somalia is thought to be more than two million people, and this does not account for the many refugees who have fled

the country altogether. Numerous organizations have documented conflict and disaster as the primary drivers of displacement for Somalis, including pastoralists [9,10].

Pastoral migration, whether voluntary or forced, is a complex phenomenon with which scientists continue to grapple. While the drivers of migrations are broadly understood, less is known about *how* environmental and conflict variables affect movement patterns across space and time in the past and how they may continue to evolve in the future. To the authors' knowledge, there is very limited public information available about pastoral populations in Somalia or the self-declared autonomous region of Somaliland. Without information that captures the dynamic nature of pastoral populations, it is extremely challenging to understand the evolving movement patterns in response to environmental changes and ongoing conflict on a regional scale. To date, several studies use computer models to explore pastoral migration through virtual environments, which will be referred to in the following paragraph. Computer models enable scenario simulation in which the relationship of variables are tested and analyzed over time and/or space, providing insight into the observed phenomenon.

Agent-based models (ABMs) are a subset of the computer modeling techniques that are increasingly utilized to examine the stochastic realities of human migration in response to environmental and conflict variables [11]. ABMs can simulate the actions of agents, such as pastoralists, based on the interactions among agents and their surrounding environment. The bottom-up nature of ABMs enables the capture of granular patterns of movement that can be summarized to a systems-wide level. While some models aim to gain a holistic understanding of the adaptive behaviors of pastoralists in response to their environment [3], others consider the possible effects of climate and/or conflict on pastoral movement [10,12,13], as well as the interplay between pastoral movement, land privatization [14,15], and disease transmission [16]. Computer models rely on a combination of data sources, qualitative and quantitative evidence, predictive datasets, and/or informed assumptions to generate simulation scenarios. The extensive spatial and temporal variability of pastoral movement and the factors that trigger it warrant the implementation of a simulation model that captures its nuances and the wide-ranging variability.

The integration of geospatial data and agent-based modeling to study pastoral migration is less explored in literature. The existing models and findings, however, have contributed substantially to a growing body of literature on the subject. Sakamoto [3] integrates low-resolution multi-temporal satellite imagery analysis and agent-based modeling to study pastoral access to resources in dryland vegetation in northeastern Nigeria, while the Center for Social Complexity and Department of Computational Social Science at George Mason University has examined the intersection of GIS and agent-based modeling through the development of a range of models on the ABM platform MASON, including the HerderLand, AfriLand, RiftLand, and RebeLand models. The models consider a range of scenarios in Eastern Africa that examine resource contention, the effect of environmental changes, availability of watering holes, and the effect of private land on pastoral movement at multiple scales [14]. While some of these models do consider ethnicity and speculative clan violence and conflict, the integration of empirical, conflict data with historical terrestrial variables such as remotely-sensed vegetation indices has yet to be broadly adopted. Our intent, influenced by the need to support the humanitarian sector which increasingly faces the complicated nexus of conflict and climate change, was to capitalize upon open-source, empirical data to build a preliminary model to better understand the additive effects of these and other variables on pastoralist populations.

Specifically, the purpose of this study is to examine the relationship between environmental change, conflict, and pastoral movement in Somaliland between 2008 to 2018 through agent-based modeling. The model generates synthetic movement patterns for nomadic pastoralists in the region, which are influenced by a series of environmental, interpersonal, and

transactional variables. Evidently, computer modeling presents high levels of uncertainty but also has immense potential to improve humanitarian preparedness and response. To improve modeling capacities, it is necessary to continue developing the input data and methodology to identify the model's successes and limitations. This paper adds to a growing body of literature that considers the use of predictive modeling and geospatial analysis to address issues in the humanitarian sector where climate and conflict variables have significant impact on population movement.

## Methods

### Setting

This agent-based model simulates the movement of nomadic pastoralists in response to conflict and environmental variability on a monthly basis in northern Somalia, specifically in the seven administrative regions of Somaliland and Puntland (Fig 1), between January 2008 and December 2018.

### Data collection and manipulation

Data utilized for this study (Table 1) were obtained from diverse sources, are of disparate typologies, and were quantified, normalized, and utilized in various ways to generate agents, assign attributes to those agents, and develop an environmental model. The following is a description of those data sources, manipulations, and the mechanism for normalization. All but one dataset were open source (S2 Supporting Information) and readily available on the internet.

**Manipulation and normalization.** The model environment consists of terrestrial variables including slope, surface water, points of artificial water sources, and vegetation rigor; interpersonal components including ethnic boundaries and locations of conflict; and a transactional variable that delineates private and public land. All data values were rasterized and aggregated to a 1 km$^2$ grid. Variables were then normalized to range values between 0–1, utilizing the equation, below.

$$Vnormalized = (Vcurrent - Vminimum) \div (Vmaximum - Vminimum)$$

Wherein *V* is the variable in consideration.

Slope was calculated from remotely-sensed elevation data obtained from DIVA GIS using the Slope Spatial Analyst in ArcMap 10.6.1. The presence or absence of surface water was ascribed a binary score of 1/0 (presence = 1) based upon data collected by the European Union Joint Data Center, downloaded through Google Earth Engine. Artificial water sources, such as wells and boreholes, were gleaned from a survey-based, geocoded dataset provided by SWA-LIM, with cells containing more than 1 water source having higher values than pixels that contained only one or none.

The vegetation score was calculated on a raster grid containing seasonal median pixel values. To calculate the median pixel values, all available MODIS satellite data was seasonally aggregated and the median pixel value was calculated. Following this calculation, the Soil-Adjusted Vegetation Index (SAVI), equation below, was applied to those median pixel values for each season in the study period to create a map of proxied vegetation availability. This index has proven more appropriate for the presence of vegetation in arid regions [17]. In this equation, *NIR* stands for near infrared values and *L* equals the canopy background adjustment factor, which was set at 0.5 to minimize soil brightness variations and eliminate the need for

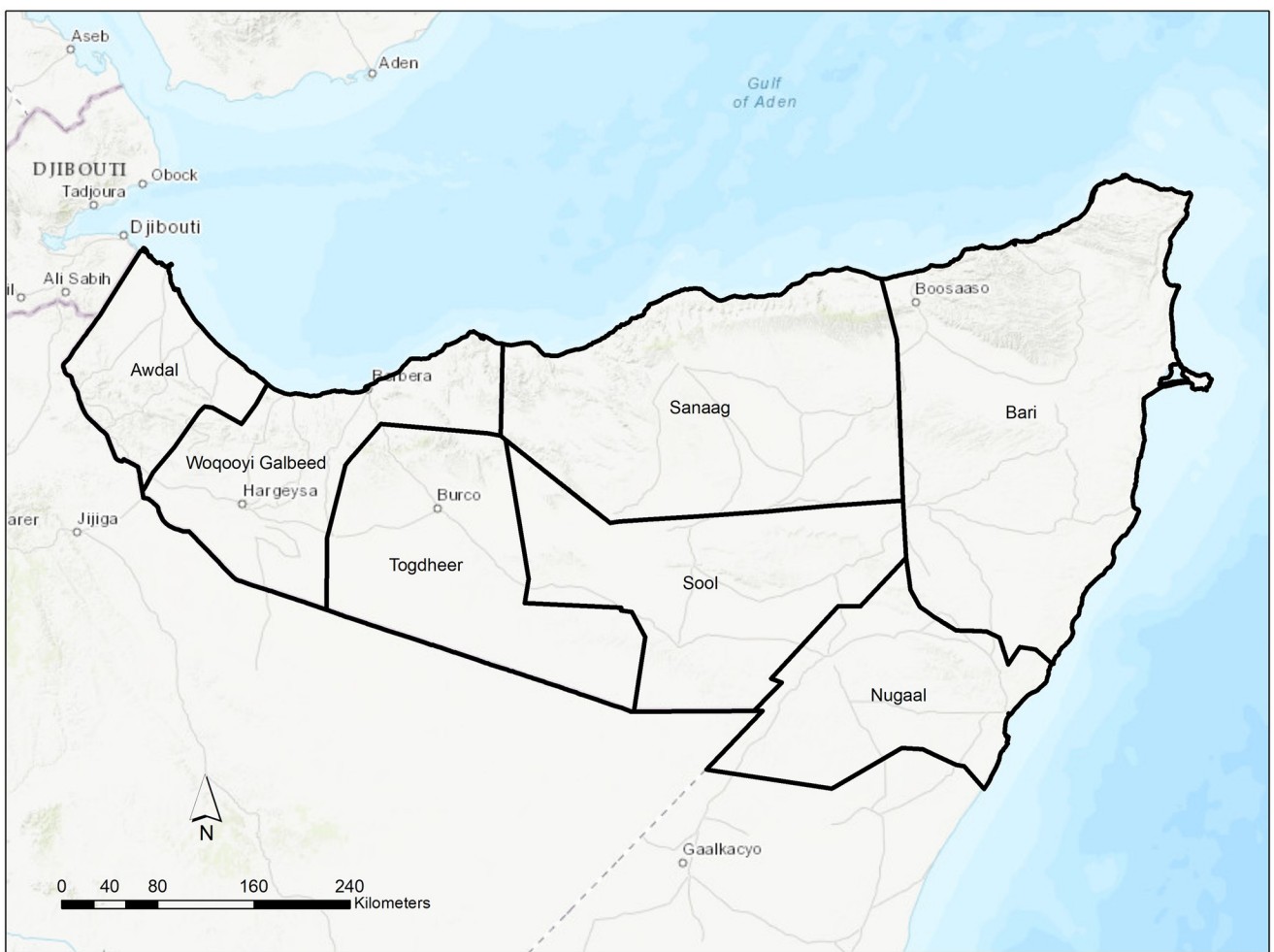

**Fig 1. Map of the boundaries of administrative 1 districts in Somaliland and Puntland.** The administrative boundaries are from the Humanitarian Data Exchange, which makes data available under the Creative Commons Foundation and the Open Data Foundation. The basemap is reprinted from ArcGIS Online under a CC BY license, with permission from ESRI, original copyright 2020 ESRI. The basemap is supported by Esri, HERE, Garmin, Intermap, increment P Corp., GEBCO, USGS, FAO, NPS, NRCAN, GeoBase, IGN, Kadaster NL, Ordnance Survey, Esri Japan, METI, Esri China (Hong Kong), (c) OpenStreetMap contributors, and the GIS User Community.

further calibrations around soil-type.

$$SAVI = \frac{(1 + L)(NIR - Red)}{(NIR + Red + L)}$$

Ethnic boundaries obtained from a static map provided by the Kenya Somalia Consortium were digitized and rasterized, and buffers were applied to the borders at 10 and 20 meters, with descending weighted values of 0.5 and 0.25 respectively. Conflict point data was gleaned from the ACLED database and rasterized. Grid cells that had higher incidence of conflict were attributed higher values, and a temporal lag was assigned within the model, with conflict occurring the previous season having half the effect on environmental favorability the following season.

In the absence of reliable data, the delineation of private and public lands was determined through expert opinion derived from a workshop convened in Nairobi, Kenya in June 2019. The workshop participants were recommended to the research team by UNICEF Somalia

**Table 1. Data utilized for the agent-based model, including ontological category, type of variable, source and date of collection.**

| Ontological Category | Computational Variable | Variable Format | Data Source | Time when data was collected |
|---|---|---|---|---|
| **Agent demographics** | | | | |
| | Population distribution | Tabular population estimates by administrative boundaries | UNFPA Population Survey Data | 2014 |
| **Environment data** | | | | |
| | Administrative boundaries | Shapefile | UN OCHA Somalia, obtained through HDX | 2018 |
| | Slope | Remotely-sensed raster layer extrapolated from elevation data | DIVA GIS | 2008 |
| | Surface water | Raster layer of remotely-sensed water sources utilized to create a NDWI Layer | NDWI Layer from the European Union Joint Research Center, obtained from Google Earth | Aggregation from 1948 and 2018 |
| | Artificial water sources | Geo-tagged data on artificial water sources | Somalia Water and Land Information Management (SWALIM) | Data from January 2008 until May 2018 |
| | Vegetation data | Raster layer of remotely-sensed data utilized to create a SAVI layer | MODIS Terra Vegetation Indices 16-Day Global | Data from 2008 until 2018 |
| | Ethnic boundaries | Polygons that delineate a general understanding of geographic ethnic boundaries | Kenya Somalia Consortium, Clan map was digitized and geocoded | Data from 2015, as recorded in 1999 |
| | Conflict data | Geo-tagged conflict data | Armed Conflict Location & Event Data Project (ACLED) | January 2008 through December 2018 |
| | Settlements (Private/public land delineation) | Polygons created utilizing buffers around townships and cities | Metropolitan data provided by the Humanitarian Data Exchange (HDX), uploaded by UNOCHA Somalia | May 2011 |
| | Land Cover | Polygon maps of land cover | Food and Agriculture Organization | May 2007 |

based on job relevance and personal connection to nomadic pastoralists in Somaliland. The participants were either based in Somalia or Somaliland, or a regional office with a focus on Somaliland. Based on local knowledge, the participant group estimated that approximately 70% to 80% of the land in Somaliland was publicly owned and that private land extended approximately 15 km beyond a large settlement and 5 km beyond smaller towns. Based on these estimations and available spatial data, the research uses the following rule: Eighty percent of land in Somaliland was estimated to be publicly-owned, and the distribution of private land was assumed to be land which extends approximately 15 kilometers from the centroid of large metropolitan centers and five kilometers from smaller settlement centers. To adhere to the 80% threshold reported, the private land boundaries around cities and towns were modified to 14 and 4 kilometers, respectively.

## The computational model

This agent-based model was developed in RePast (Recursive Porous Agent Simulation Toolkit) Simphony 2.6 using a Java-based simulation environment. Repast is a leading open-source ABM development toolkit specifically designed for social science applications and has been well regarded in comparison to other ABM platforms [18]. The Repast development framework provides all the basic functionality required to support the execution of an ABM, including scheduling mechanisms and diverse modeling functions. The established framework enables researchers to add components to customize the model to fit their needs, allowing the environment to be modified and to incorporate a range of dynamic variables. The computations in this paper were run on FASRC Cannon Cluster supported by the FAS Division of Science Research Computing Cluster at Harvard University.

This agent-based model includes two entities: 1) agents, each representing a single nomadic pastoralist household unit, and 2) the physical environment, which is a geospatial landscape composed of both dynamic and static attributes. At the start of the simulation, agents are synthesized for the simulation. Each generated agent is assigned attributes, including their geographic position at the start of the simulation, the name of the administrative unit they fall within, their ethnicity, and clan association. The number of agents generated per administrative unit was informed by a Population Estimation Survey conducted in 2014 [19]. Within any given administrative unit, the agent start position was randomly generated with two constraints: 1) the agent must not be in unsuitable landscapes including water bodies and areas of bare soil (i.e. sand), and 2) the agent must be located 14 kilometers or more outside a major city and four kilometers or more outside of smaller settlements, i.e. on 'public land'. For every simulation run, the agent start position remains identical.

The gridded physical environment is composed of 1 km$^2$ grid cells and has a spatial extent of approximately 490,000 km$^2$, which covers the administrative regions in Somaliland and Puntland. The environment was designed to include eight variables, grouped into three thematic components: terrestrial variables, interpersonal variables, and transactional variables (Table 2). The environment variables are categorized to be either pull (attractors) or push (detractors) factors for nomadic pastoralists whose patterns of movement are influenced by the availability of water and suitable grazing land to support their herd. The factors were identified and included based on literature, workshops, and/or discussions with regional experts. Intuitively, the presence of vegetation (as proxied by SAVI) and availability of water are considered attractors, while steeper terrestrial slope and proximity to conflict or potential ethnic tension (as proxied by proximity to ethnic borders) are designated detractors.

Variables such as surface water availability, conflicts, and vegetation cover are subject to seasonal changes. In this model, four seasons exist per annum. The dry season from December to March is locally referred to as *Jilaal*, which is followed by the long rainy season, *Gu*, from April to June. The dry season that follows, *Hagaa*, spans from July to September while the short rainy season that takes place between October and November is known as *Deyr*. The surface water layer is disabled during dry seasons, as nomadic pastoralists tend to rely on manmade water sources such as wells and boreholes during these periods and when surface water is extremely sparse. The conflict environmental surface changes seasonally during the model's timeline, with conflict point data being aggregated to each season and changing at the end of the ascribed three-month period. Vegetation availability scores also change seasonally, based upon the medial pixel value of the imagery available for each defined season, as described

**Table 2. Attributable variables that are considered in the environment, with favorability score, relative impact and change status.**

| Thematic Variables | Attributable Variable | Impact on Favorability | Change Over Time |
|---|---|---|---|
| Terrestrial | Gradient of the land (slope) | The greater the gradient, the less favorable | Static |
| | Surface water | Proximity to water source is favorable | Changes seasonally, only enabled in wet seasons |
| | Artificial water sources | Proximity to water source is favorable | Static |
| | Vegetation data | Higher SAVI score correlates with more vegetation, which is favorable | Changes seasonally |
| Interpersonal | Ethnic boundaries | Proximity to ethnic boundaries is less favorable (with a gradient buffer of 20km) | Static |
| | Conflict data | Proximity to conflict is less favorable (no gradient buffer) | Changes seasonally |
| Transactional | Private/public land ownership | Private land requires a transaction between the land-holder and the pastoralist to establish land-sharing. If no agreement is made, the pastoralist must find another land parcel | Static |

above, with applied SAVI scores. The inclusion of these real, longitudinal data reflect not only the seasonal changes associated with wet and dry seasons, but also the changing environment due to climate and human variables over the course of the simulation.

The favorability score for each land parcel is artificially created through the additive equation:

$$Score_{pixel} = v1 + v2 + v3 - (0.25 * v4) - v5 - (0.25 * v6)$$

Wherein,

v1 = normalized vegetative cover, as calculated with SAVI, where L = 0.5

v2 = normalized surface water index (enabled in wet seasons)

v3 = artificial water point sources

v4 = normalized terrain gradient

v5 = normalized conflict frequency

v6 = ethnic boundary

**Simulating movement.** Agents move throughout the simulated geographic environment based on the fundamental premise that nomadic pastoralists rely on livestock to support their livelihoods, and are therefore heavily dependent upon access to water and vegetation (Fig 2). At every time tick, i.e. one month, the agent searches for a cell in the environmental grid with the highest favorability score within a search radius (referred to as scouting range) of its surrounding environment. This scouting range is a random distance generated between a 15 and 30 kilometer radius, based on expert consensus regarding the monthly mobility capacity of nomadic pastoralists. Once the most favorable cell has been identified, the agent moves to this location and determines whether it is located on private or public land. If the cell is public land, they are free to move to that location without any further delay and have the option to stay there for the duration of the season (between one and three months), provided this location continues to have the best favorability score. Once the season changes, the agent is required to seek out further land. This latter constraint is an effort to model resource depletion of that grid given the grazing requirements of a pastoralist herd.

However, if the cell happens to be on private land, the agent must negotiate a land-sharing deal with the local landowners. If the agent successfully makes a deal with the landowner, they are able to remain at that location for the duration of the season. However, if the agent is unable to obtain access, they are required to move to another grid cell within the same time tick, consider whether the new grid location is private or public land, and repeat the process described above. This secondary selection of a cell is determined by the next best score in the gridded environment.

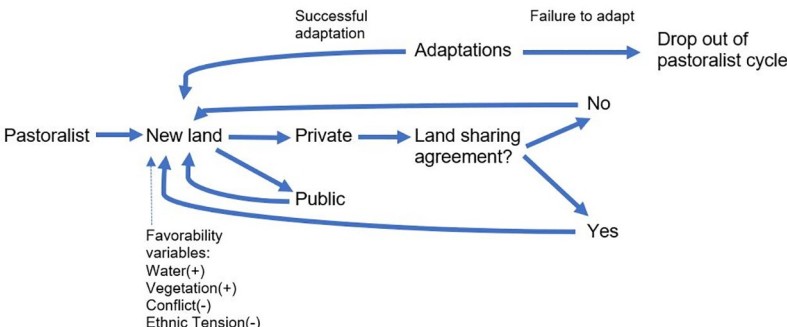

**Fig 2. Conceptual model for the ABM of nomadic pastoralist agents seeking new grazing land.**

**Table 3. Number of synthetic agents generated per administrative unit for the simulations.**

| Administrative Unit | Number of Agents (10% of admin pop) | Number of Agents (25% of admin pop) | Number of Agents (75% of admin pop) |
|---|---|---|---|
| Awdal | 2,851 | 7,128 | 21,383 |
| Woqooyi Galbeed | 4,374 | 10,935 | 32,806 |
| Togdheer | 2,428 | 6,071 | 18,214 |
| Sool | 2,898 | 7,246 | 21,739 |
| Sanaag | 4,776 | 11,941 | 35,823 |
| Bari | 1,911 | 4,779 | 14,336 |
| Nugaal | 3,337 | 8,342 | 25,025 |
| Total | 22,575 | 56,442 | 169,325 |

If the agent is unable to make a deal on three separate occasions in any given season, the agent state switches from pastoralist to IDP, at which point the agent exits the simulation or 'drops out'. This is done with the assumption that the lack of access to grazing land leads to the death of livestock, and the pastoralist agent is required to seek alternative livelihoods and will potentially drop out of a purely pastoralist lifestyle.

The model was run for eleven years, from January 2008 to December 2018, with one-month change and decisional intervals. An explanation of the model is described in a structured Overview, Design Concepts, Details protocol, which has been amended to include details regarding human decision-making (ODD+D) [20], and can be found in S3 Supporting Information. Given the constraints of computational power, completing the simulation for all 225,767 pastoralist agents was not possible.

Thus, a random sampling of 10%, 25%, and 75% of the population in each administrative region was conducted, and the simulation was executed with this subset of the generated agents (Table 3). The spatiotemporal outputs generated by these three sample thresholds were compared to each other to capture similarities and differences between the sample sizes. The comparisons revealed that the 10% sampling threshold generated similar results to the simulations using a 25% and 75% agent sample (Fig 3). Fig 3 shows that the agent population declines in a similar manner in all administrative regions across the sampling thresholds January 2008 and October 2018 and display less than a 1% variation between iterations. This internal validation process confirmed that the use of a random sample at 10% would sufficiently reflect the population in the study area. Subsequently, three additional simulation iterations were produced using a 10% sample size and results were cross compared. Again, each iteration produced similar results with percent population per district over time varying less than 1.2% throughout all iterations at the four, previously utilized time points, thereby justifying the use of one 10% run for future spatiotemporal analysis.

## Analysis

Spatial analysis of this agent-based model is preliminarily aimed at identifying temporal trends of pastoralist density both across the seasons of a year and between seasons influenced by a decade of evolving conflict and climate-dependent terrestrial variables. Somalia experiences four distinct seasons each year, as described above. Subsequently, the months of January, May, August, and October were identified as representative midpoints of these seasons and utilized in this spatial analysis.

By analyzing the population distribution of pastoralist agents resulting from the ABM simulation during each of these seasonal time points through choropleth maps of population counts, kernel density maps (KDM), and standard deviational ellipses analysis (SDE), a

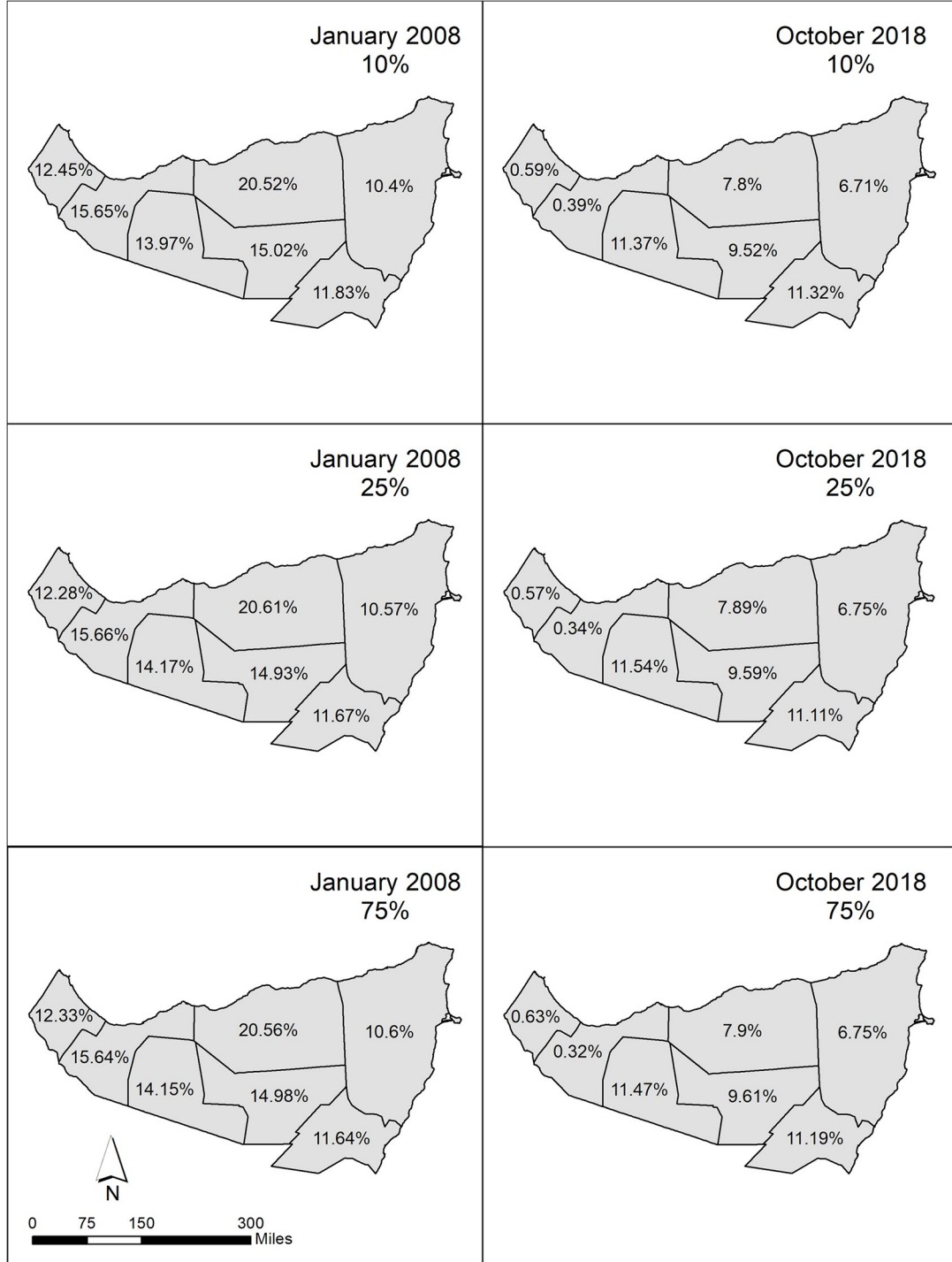

**Fig 3. Proportional agent population at January 2008 and October 2018 considering a 10%, 25%, and 75% sampling of the agent population.** The administrative boundaries are from the Humanitarian Data Exchange, which makes data available under the Creative Commons Foundation and the Open Data Foundation.

spatio-temporal understanding of movement is created based upon what can be considered 'natural' seasonal dynamics. The comparison of population distribution during specific seasonal periods from 2008 to 2018 heightens the analysis to appreciate how seasonal migration patterns have changed over the decade. The spatial analysis presented in this paper compares the start year, 2008, and the end year, 2018, to capture total change over time. The juxtaposition of these outcomes with environmental variables, such as artificial water sources, vegetation cover, and conflict allows for associations regarding the interplay between changes in resource availability, conflict dynamics and pastoralist movements.

All spatial data was projected in the World Geodetic System (WGS) 1984 Universal Transverse Mercator (UTM) Zone 38N and all analysis was performed utilizing the ArcGIS 10.7 platform by ESRI.

**Population counts and differences.**   Population counts were created by identifying the location of each pastoralist agent at the aforementioned time periods and summed to generate aggregate population counts across the study region. These data were then spatially joined with administrative districts within Somaliland and Puntland. Choropleth maps delineating population counts per administrative district were created excluding those agents that lay outside of boundary lines. Classification was done using Jenks Natural Breaks without justification between the two time periods, as justification would obscure the significant difference in population counts in each district. Differences between the January 2008 population counts and January 2018 and October 2018 counts were calculated to create a change layer. These calculations were undertaken to explore the change in population counts between 2008 and 2018, both taking into account season (January 2008 versus January 2018) and the beginning and end of the simulation.

**Kernel density maps.**   Kernel density maps create a predictive distribution surface map of pastoralist populations given known spatial sample inputs. Population density is calculated using the quadratic formula below, with the highest weight ascribed to the known point location and tapering to zero at the edges of the search radius and the predicted population at any given cell in the output raster map being an accumulation of the values for each of the calculated surfaces. Thus,

$$Density = \frac{1}{(radius)^2} \sum_{i=1}^{n} \left[ \frac{3}{n} \times pop_i \left( 1 - \left( \frac{dist_i}{radius} \right)^2 \right)^2 \right]$$

wherein: $i = 1,...,n$ are the input points within the search radius; $pop_i$ is the population field value of point I, and $dist_i$ is the distance between point i and the (x,y) location [21]. The cell size (x,y) is 0.0089831528, 0.0089831528 decimal degrees, which is equivalent to 1 km². Given the UTM projection of the environmental basemap, the analysis was parameterized to a planar method of measurement. By creating these kernel density maps of the pastoralist population over each of the four seasons of 2008 and 2018, a spatio-visual interpretation of how the pastoralist population has evolved over the course of ten years and its seasonal permutations may be seen.

**Standard deviational ellipses.**   An unweighted, standard deviational ellipse analysis was undertaken to characterize the spatial distribution—specifically the location, dispersion and orientation [22]–of pastoralists over time, utilizing classical statistical methods with the following equations:

$$C = (var(x) \; cov\,(x,y) \; cov\,(y,x) \; var(y)) = \frac{1}{n} \left( \sum_{i=1}^{n} \tilde{x}_i^2 \; \sum_{i=1}^{n} \tilde{x}_i \tilde{y}_i \; \sum_{i=1}^{n} \tilde{x}_i \tilde{y}_i \; \sum_{i=1}^{n} \tilde{y}_i^2 \right)$$

where:

$$var(x) = \frac{1}{n}\sum_{i=1}^{n}(x_i - \underline{x})^2 = \frac{1}{n}\sum_{i=1}^{n}\tilde{x}_i^2$$

$$cov(x, y) = \frac{1}{n}\sum_{i=1}^{n}(x_i - \underline{x})(y_i - \underline{y}) = \frac{1}{n}\sum_{i=1}^{n}\tilde{x}_i\tilde{y}_i$$

$$var(y) = \frac{1}{n}\sum_{i=1}^{n}(y_i - \underline{y})^2 = \frac{1}{n}\sum_{i=1}^{n}\tilde{y}_i^2$$

and $x$ and $y$ are the coordinates for each pastoralist at each defined time-point $i$, $\{\bar{x}, \bar{y}\}$ represents the mean center for the features, and $n$ represents the total number of features [23]. Given Rayleigh distribution, the two standard deviations applied includes 98 percent of the features. Comparing the size, shape and orientation of SDEs over four different seasons in 2008 and 2018 allows for the detection of relationships between environmental variables and pastoralist distribution not necessarily captured by density mapping alone. While SDEs have been utilized in the past to explore the relationships between environment and criminal activity [24,25], to characterize racial segregation [26], and to assist in outbreak surveillance [27], the application of SDEs to ABM outputs, specifically in the context of pastoralist migration, is novel.

## Results

The results of the ABM yielded the location of each pastoralist agent at every time step, along with its scouting range, and the favorability score of the grid that it inhabited. The following table (Table 4) quantifies the counts of pastoralist agents within each administrative boundary at each seasonal time-point during the periods chosen for analysis. In January 2008, Sanaag and Woqooyi Galbeed districts have the highest number of pastoralists, with 4633 and 3534, respectively, but all districts had populations greater than 2000. In October 2018, after nearly eleven years of simulated environmental change, the districts of Togdheer and Nugaal, with 2568 and 2556 pastoralist agents respectively (Fig 4), have the highest populations, and there is a compelling lack of pastoralists in Woqooyi Galbeed with only 89 inhabitants. In aggregate, there was a 45% decrease in the pastoralist population during the simulation period, equating to a 'drop out' of nearly 1,000 pastoralist agents per year.

**Table 4. Pastoralist agent counts per administrative district over time, a 10% sampling.**

| Administrative District | Pastoralist Population Counts | | | | | | | | Total Difference |
| --- | --- | --- | --- | --- | --- | --- | --- | --- | --- |
| | 2008 | | | | 2018 | | | | |
| | Jan | May | Aug | Oct | Jan | May | Aug | Oct | |
| Bari | 2348 | 2485 | 2425 | 2402 | 1561 | 1549 | 1518 | 1516 | -832 |
| Nugaal | 2670 | 2554 | 2526 | 2535 | 2564 | 2570 | 2563 | 2556 | -114 |
| Sanaag | 4633 | 4068 | 3741 | 3581 | 1778 | 1783 | 1761 | 1762 | -2871 |
| Sool | 3392 | 4172 | 4267 | 4346 | 2514 | 2408 | 2264 | 2149 | -1243 |
| Togdheer | 3154 | 3754 | 3779 | 3777 | 2381 | 2431 | 2470 | 2568 | -586 |
| Woqooyi Galbeed | 3534 | 2888 | 2086 | 2064 | 94 | 99 | 90 | 89 | -3445 |
| Awdal | 2810 | 2483 | 2014 | 1852 | 150 | 141 | 137 | 133 | -2677 |
| Total (Includes those that fall outside of specific districts) | 22,575 | 22,461 | 20,856 | 20,633 | 11,049 | 10,995 | 10,807 | 10,786 | -10,104 |

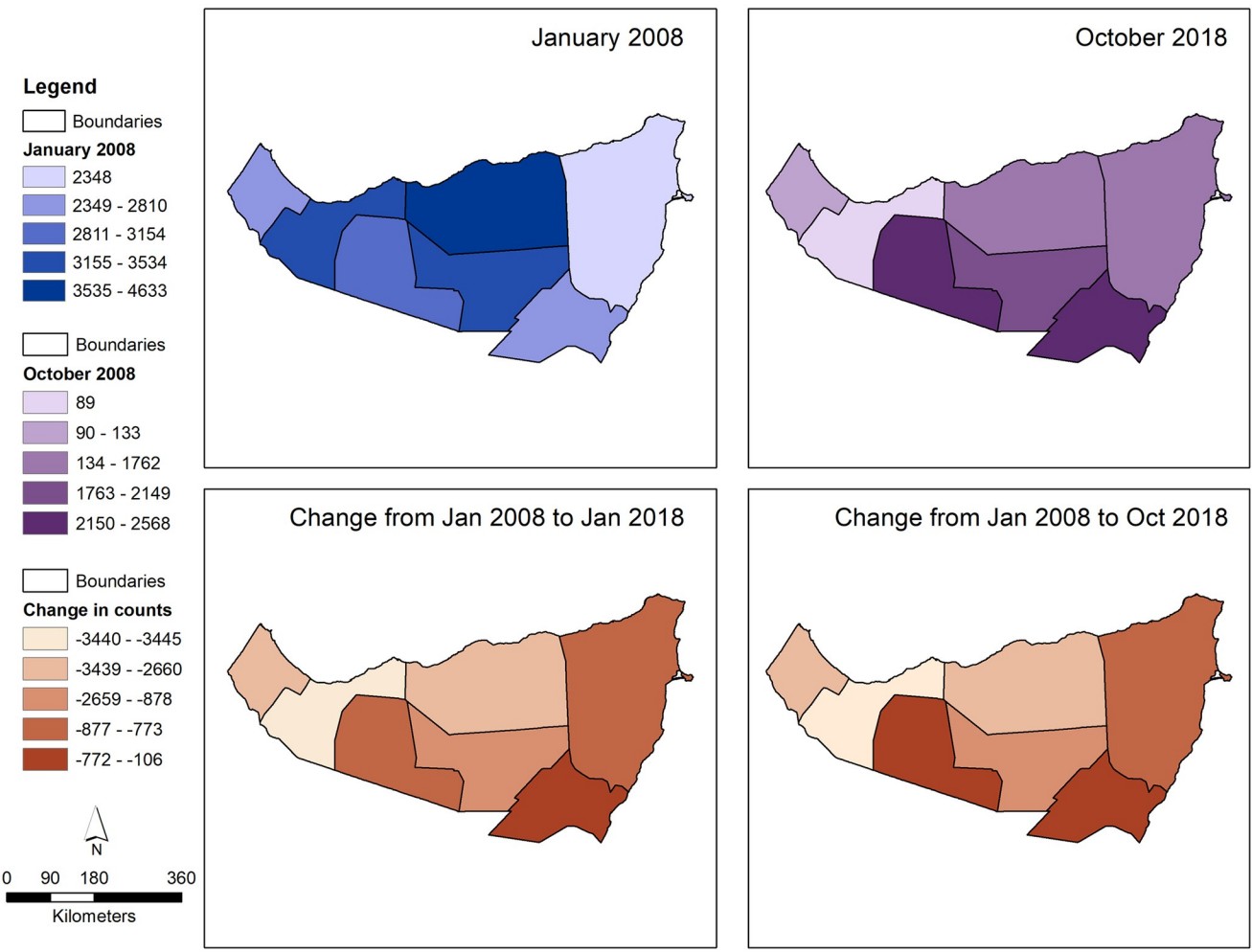

**Fig 4. Choropleth maps of pastoralist agent population count and count difference at each time point as defined above.** The administrative boundaries are from the Humanitarian Data Exchange, which makes data available under the Creative Commons Foundation and the Open Data Foundation.

All districts portray a decline in pastoralist population counts, a result of pastoralist agents 'dropping out' of purely pastoralist cycles or migrating to alternate districts (Fig 3). However, certain districts have significantly greater declines in population than others, with Nugaal demonstrating only a decrease of 114 pastoralists, and Woqooyi Galbeed having a decline of almost 3,500. Also notable is the difference in population counts between administrative districts over time. In January 2008, the difference in population count between the most (Sanaag) and least (Bari) populated districts was 2,285, as opposed to October 2018, in which the difference between the most (Togdheer) and least (Woqooyi Galbeed) populated districts was 2,479. In general, the greatest decline in population per district was noted in the northeast of the study region with very little change in this pattern when accounting for seasonal variability between January and October of 2018.

Kernel density (KD) analysis for the four seasonal time points in both 2008 and 2018 demonstrates pastoralist population density throughout Somaliland and Puntland as simulated by this ABM (Fig 5A and 5B) with statistical characteristics identified in Table 5. In January 2008, the density across the study area ranged from 0 to 0.58 agents per square-kilometer with a

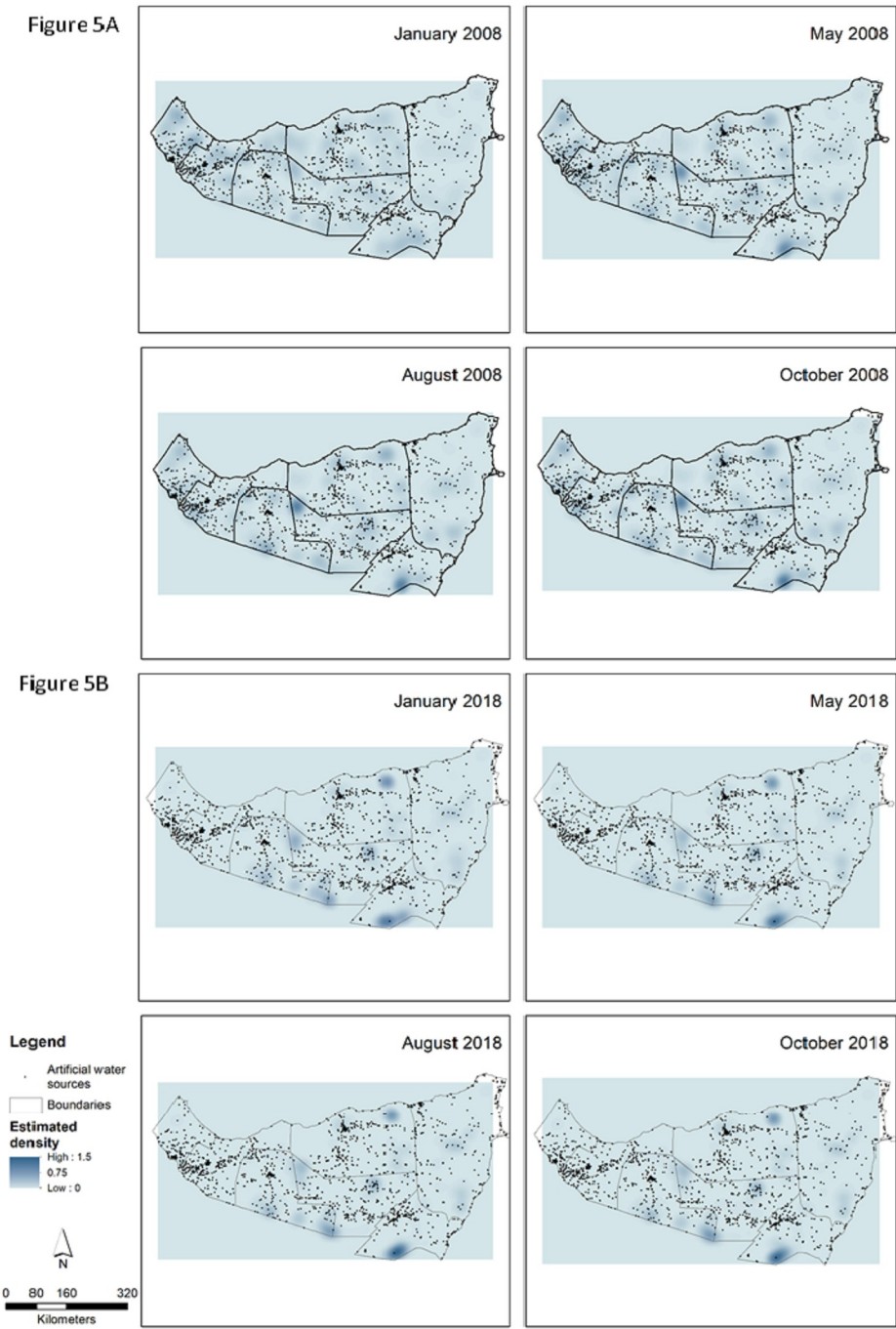

**Fig 5. A and B: Kernel density maps of pastoralist population agent positions at ascribed seasonal time points as determined by the ABM simulation for 2008 (A) and 2018 (B).** Density unit classifications have been justified across all analyses. The administrative boundaries are from the Humanitarian Data Exchange, which makes data available under the Creative Commons Foundation and the Open Data Foundation.

standard deviation of 0.73. The map in Fig 5A shows that the agent density is fairly dispersed, but the highest densities are found in Awdal, the westernmost administrative region, with the lowest densities in Bari, the easternmost administrative region. It is important to consider that January 2008 is the first month of the simulation, directly before which the agents were

**Table 5. Statistics of pastoralist population kernel density maps disaggregated by time point.**

| Month/Year | Minimum | Maximum | Mean | Standard deviation |
|---|---|---|---|---|
| January 2008 | 0 | 0.58 | 0.050 | 0.073 |
| May | 0 | 0.95 | 0.048 | 0.086 |
| August | 0 | 1.16 | 0.047 | 0.093 |
| October | 0 | 1.08 | 0.044 | 0.09 |
| January 2018 | 0 | 1.11 | 0.025 | 0.078 |
| May | 0 | 1.28 | 0.025 | 0.08 |
| August | 0 | 1.47 | 0.025 | 0.086 |
| October | 0 | 1.38 | 0.024 | 0.083 |

randomly generated in each administrative region, so the spatial dispersion is likely a remnant of the agent generation.

Four simulated months later, in the following May, the maximum population density increased to 0.95 agents per square kilometer, without a large change in standard deviation. Certain high density areas noted in January 2008, specifically in the south of Nugaal and the northwest corner of Sool, persist as relatively dense areas throughout the remainder of the simulation (Fig 5A). By August 2008, maximum density continues to increase, and additional clusters begin to form in Togdheer along the Ethiopian border as well as near the coast in Sanaag. Towards the end of 2008, high- and low-density locations appear to have reached homeostasis, with both large scale spatial patterns and standard deviations of population density remaining similar.

By January 2018, the mean density of pastoralist agents has halved (0.025 from 0.050), a result of pastoralist 'drop out'. Areas of highest population density are now centralized predominantly within four districts: Sanaag, Togdheer, Sool and Nugaal (Fig 5B). Certain early areas of pastoralist accumulation, such as that in the south of Sool and along the Ethiopian border, persist as the most densely populated. In general, as the simulation progresses, pastoralists appear to tend towards clustering, with maximum densities at 1.47 and 1.38 pastoralists per 1km$^2$ in the final two seasons. These areas of high density seem to shift eastwards over the eleven-year period, or inversely, the density of pastoralists in the western districts of Awdal and Woqooyi Galbeed declined significantly.

These density maps have been overlaid with artificial water sources that consists of wells, boreholes, dams, and other artificial facilities. The artificial water sources were included in the ABM and were one of the pull factors in the simulation. Sanaag, Sool, and Woqooyi Galbeed, the areas in which high numbers of water points are situated, understandably evidenced higher densities of agent concentration during 2008. However, this association was weakened when appreciating population density in 2018. Some relationship could be intimated between high density areas and artificial water sites in northern parts of Sool and Sanaag, but as of January 2018, there are no high density clusters in the westernmost part of the country despite the presence of artificial water sources.

Standard deviational ellipses (SDE) of the simulated pastoralist population were produced for four time points in 2008 and 2018, resulting in the creation of eight SDEs (Fig 6 with statistical descriptions in Tables 6 and 7). In 2008, there was minimal variation in the SDE area between all four seasonal time points, which ranged from 578,675 km$^2$ to 600,534 km$^2$ with a mean area of 594,767 km$^2$ and a difference of 21,859 km$^2$. The centroid of the ellipses experienced a minor shift in a Southeastern direction towards the districts of Nugaal, Sool, and Togdheer. Although the decrease in SDE area seen towards the end of the year evidences a trend towards consolidation in the pastoralist population, the relatively consistent rotation

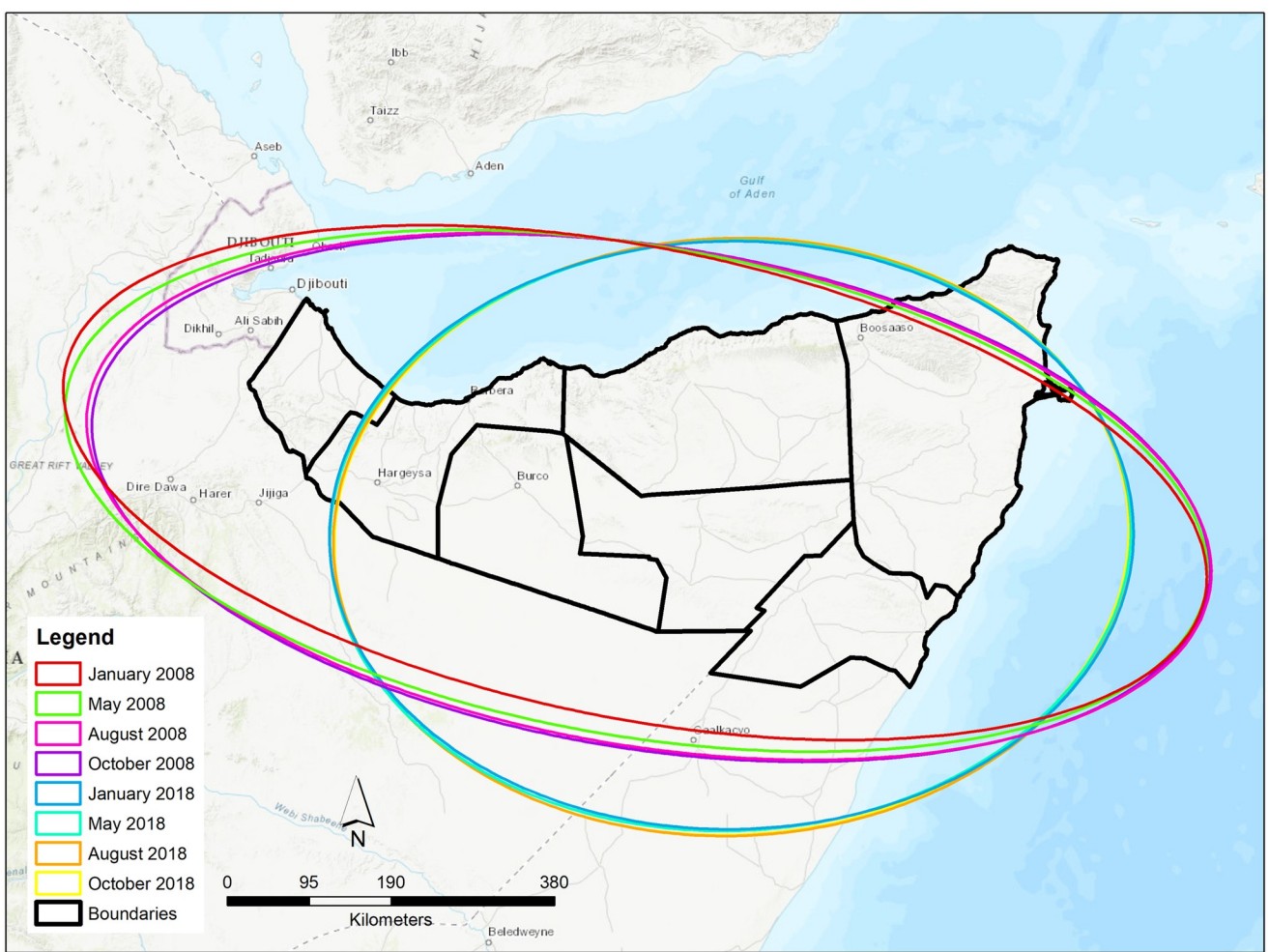

**Fig 6. Pastoralist population distribution as demonstrated by standard deviational ellipses, disaggregated seasonally for 2008 and 2018.** Two standard deviations were applied to capture 98% of features. The basemap is reprinted from ArcGIS Online under a CC BY license, with permission from ESRI, original copyright 2020 ESRI. Basemap contributors include Esri, HERE, Garmin, Intermap, increment P Corp., GEBCO, USGS, FAO, NPS, NRCAN, GeoBase, IGN, Kadaster NL, Ordnance Survey, Esri Japan, METI, Esri China (Hong Kong), (c) OpenStreetMap contributors, and the GIS User Community.

angle indicates little to no change in the spatial orientation of the pastoralist population across seasons in 2008.

In 2018, the area of the SDE ranged from 500,315 km$^2$ to 504,424 km$^2$ (Table 7) with a mean area of 501,781 km$^2$ and a difference of only 4,109 km$^2$. The centroid of the SDE moved slightly between the seasons and, similar to 2008, the rotation angle only varied by approximately two degrees. As demonstrated in Fig 6, the SDEs calculated for 2018 largely exclude the

**Table 6. Characteristics of standard deviational ellipses in 2008.**

| Month | Area km$^2$ | Centroid (X,Y) in meters | Rotation angle |
|---|---|---|---|
| January 2008 | 578,675 | 694,751.22, 1,057,483.83 | 101.28 |
| May 2008 | 599,409 | 696,320.09, 1,048,051.87 | 99.83 |
| August 2008 | 600,534 | 710,984.51, 1,041,396.29 | 99.47 |
| October 2008 | 600,450 | 713,028.55, 1,039,774.32 | 99.11 |

**Table 7. Characteristics of standard deviational ellipses in 2018.**

| Month | Area km$^2$ | Centroid (X,Y) in meters | Rotation angle |
|---|---|---|---|
| January 2018 | 500,315 | 806753.97, 996,614.78 | 89.37 |
| May 2018 | 500,475 | 805,644.60, 996,091.99 | 88.12 |
| August 2018 | 504,424 | 807,421.15, 994,493.52 | 88.15 |
| October 2018 | 501,910 | 806,805.26, 995,034.55 | 87.48 |

western district of Awdal, indicating a statistically significant decrease in pastoralists in that region in comparison to 2008. When comparing the 2008 and 2018 standard deviational ellipses, these results support a previous analysis that the overall grazing patterns of pastoralists have become more compact, with a mean SDE areal decrease of 92,986 km$^2$. Similar to the outcomes of the kernel density maps, the SDEs demonstrate a shift in pastoralist density towards the southeast. And while there is minimal seasonal variability within years, it is notable that SDE areal differences between seasons were higher in 2008 at 21,859 km$^2$ in comparison to only 4,109 km$^2$ in 2018, evidencing a decrease in spatial variability towards the end of the model. It is important to note that this analysis does not account for the variability within the 11-year period and only considers the beginning and end years to capture the absolute change over time.

## Population, SAVI, and conflict trends

A direct, temporal comparison between pastoralist population size generated by the ABM, conflict incidence, and SAVI per district are illustrated in Fig 7. In general, population size shows negative trends over the time period, intimating either outmigration or drop-out of the agent from the pastoralist cycle. Soil-adjusted vegetation indices vary similarly across the region over time, with districts largely maintaining a set hierarchy regarding relative vegetation availability. Incidents of conflict vary considerably between districts and over time, but Bari overshadowed other districts, with nearly 100 incidents of conflict in early 2018 –more than double the highest seasonal incidence in the second most conflict-affected district of Togdheer.

Looking across the population, vegetation, and conflict trends reveals compelling associations. Notably, the three districts with the lowest SAVI scores between 2008 and 2018, Awdal, Sanaag, and Woqooyi Galbeed, show the steepest rates of attrition during the simulation. Conversely, the districts of Sool and Togdheer show relatively slower population depletion, and even exhibit early increases in population size in the beginning of the simulation. Both of these districts possessed higher SAVI scores than neighboring districts with the highest attrition rates. While Bari is adjacent to Sanaag, and possesses a higher SAVI score than Sool, the number of incidents of conflict are significantly higher, clearly dampening the favorability of migration into that district. Nugaal, in the Southeastern most part of Puntland, displayed the steadiest population size over the eleven years, which is likely attributable to high vegetation indices, low incidence of conflict, and relative geographical isolation. In the first three months of 2010, Sool experienced 22 incidents of conflict, which likely lead to the efflux of pastoralists into neighboring Togdheer, as seen in Fig 7a. Similar patterns are seen at the beginning of 2015 and between March and June of 2018, when Sool's conflict incidents exceeded Togdheer's (26 to 18 and 42 to 7, respectively). The inverse, however, is not evidenced. For example, a significant spike in conflict in early 2016 in Togdheer did not result in a large flow of pastoralist agents into Sool as would be expected.

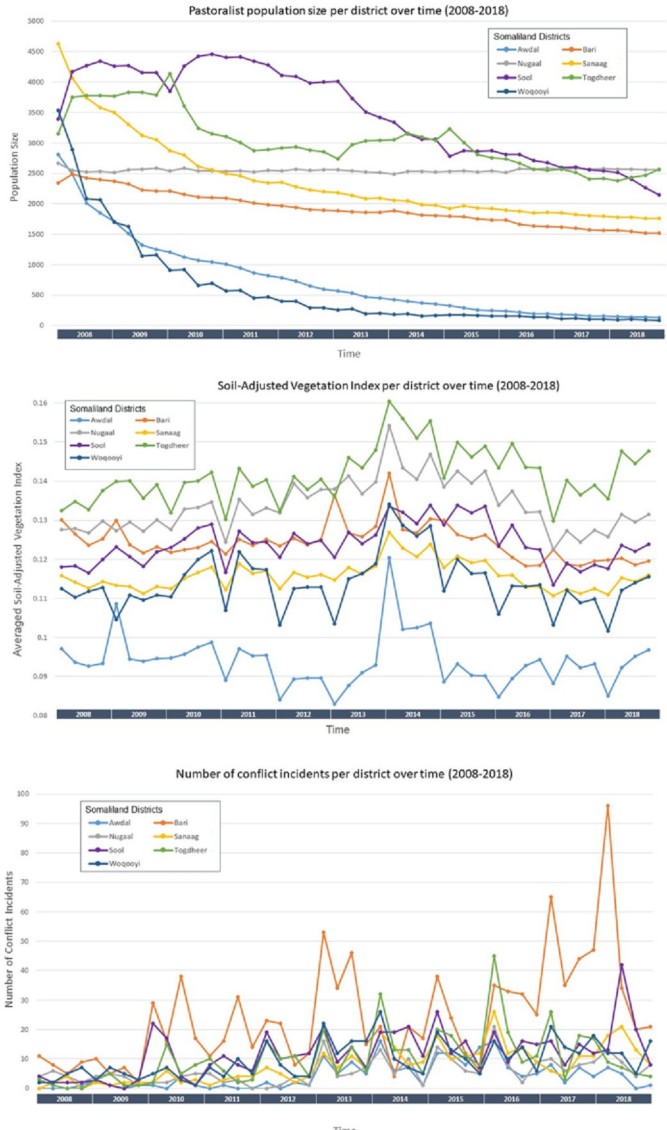

**Fig 7. Pastoralist population size, soil-adjusted vegetation index, and number of conflict incidents per district over time between January 2008 and December 2018 as aggregated to season.**

## Discussion

In general, this research demonstrates compelling trends of consolidation, clustering, and a large-scale spatial movement of agent density in response to terrestrial, conflict, and transactional variables. Clearly, soil-adjusted vegetation as a proxy for vegetation availability significantly influenced pastoralist migration, with both Awdal's and Wooqoyi Galbeed's chronically low SAVI scores likely driving the significant egress of pastoralists out of this region despite relatively low conflict indices. However, this relationship is clearly not that simplistic, given that the districts that are highest in the vegetation availability hierarchy (e.g. Togdheer and Nugaal) do not exhibit significant influx, as the pull effect of SAVI is likely dampened by alternate variables such as topology, conflict, and geographic isolation. This speaks to the complexities of the model and the interactions between the environmental, conflict, and transactional

variables, which allows for deeper, more cogent narratives regarding this nexus to be derived with closer inspection.

For example, between April and December of 2016, Somalia experienced a significant drought and tremendous food insecurity [28]. This is reflected in the significant decreases in SAVI captured in most districts during 2016 (Fig 7). The districts of Sanaag and Bari were relatively spared (with comparatively less extreme declines in SAVI at that time), however, no significant increase in pastoralist population size occurred. While it is feasible that other variables such as ethnic boundaries or topography played a role in agents' decisions around migration, a conceivable explanatory factor may be the sharp up-tick in conflict in those districts at the beginning of 2016.

The relatively steady pastoralist population seen in Bari over the entirety of the study period is also compelling, given the disproportionately high incidence of conflict in the region. One would assume that the push impact of conflict may have been offset by comparative vegetation availability, however, the neighboring districts of Nugaal and Sool have similar if not higher SAVI values and far lower conflict rates. And on a smaller spatial scale, the patterns of small yet persistent areas of high pastoralist density (Fig 5), as seen in the south of Nugaal, the northwest horn of Sool, and the northeast of Sanaag, are intriguing. The positioning of artificial water source did appear to yield some influence over pastoralist density in 2008, but these associations largely resolved by the end of the model period and likely do not account for the persistence of these densities. A more nuanced interrogation of the environmental favorability surface could proffer further understanding of the heterogenous way in which terrestrial, conflict, and transactional variables influence pastoralist dynamics.

This model resulted in minimal seasonal variation of pastoralist population distribution during the study period. This outcome is unanticipated, given that long-distance movement is a coping mechanism during dry periods when water and vegetation are scarcer [28–30]. It is feasible that the inclusion of water and vegetation proxy indicators into a simple, non-weighted, favorability score masked any potential effects at this time scale. However, another explanation is present in the spatiotemporal SAVI data, which reveals that the vegetation indices varies on a larger spatial scale, often affecting the whole of study area similarly and not disrupting the hierarchy of vegetation availability. While there are time periods when districts have minor rearrangement in this hierarchy, it is possible that conflict, distance, topography, ethnic boundaries, and other variables minimize the pull factor of such a small seasonal change. Notwithstanding, further discussion and validation of the weight of these variables in the decision to migrate needs to be undertaken to better model any presence or absence of seasonal movement.

This study incorporates several novel components into the field of agent-based modeling for migration research. First is overarching methodology that utilizes open-source data to create an agent-based model superimposed on a real, geographic landscape, and the subsequent exploratory spatial analysis that allows for hypothesis creation and plausible narratives regarding the nexus of conflict, climate, and migration. Second is the utilization of SAVI to more accurately capture the vegetative cover of Somaliland and Puntland. While Normalized Difference Vegetation Indices (NDVI) have been included in the past [3], the adoption of SAVI as an indicator is likely more accurate in arid landscapes. Second is the incorporation of empirical, historical conflict data.

Third is the mechanism by which agents were exposed to dependent variables and made decisions. Typically, ABM studies have considered the incorporation of variables one by one, where the agent makes a decision based on a decision tree. This study, however, normalized and aggregated all independent variables to a single favorability raster surface. While there is one favorability score generated, this model allows for the data layers to be weighted in a

variety of combinations to understand the effects of individual variables rather than the collective whole.

In building the conceptual model, we invoked a number of disciplines for our assumptions. Social science, climatology, conflict studies and spatial epidemiology all contributed to its development. Such a multi-disciplinary approach is required to understand real-world complex ecosystems and for us, to ideally produce data that would allow us to create inferences between change in terrestrial, interpersonal and transactional variables and population movement and density. We believe the model proposes a tool by which the spatially-relevant critical domains that drive migration—climate and conflict factors in particular—can assist the global and local humanitarian governmental and non-governmental organizations that work to support livelihoods in populations affected by protracted conflict to anticipate and prepare for these effects. To that end, while this model was a proof of concept and simulates an agent population based on limited empirical data, fine-tuning this model by improving its precision through in-depth qualitative, exploratory data on pastoralist behavior and decision-making would further engage the global humanitarian community in ways beyond their current analytic capacity.

## Limitations and challenges

The model in theory includes a large number of agents making monthly decisions based on the influence of eight variables over a 11-years period resulting in hundreds of thousands of potential data points, requiring a vast amount of computing power. For us, this is limiting for two reasons: first, it required us to use a sample of agents, introducing some degree of statistical error that we could not determine after running the model; second, to demonstrate the value of the proof-of-concept model, the team chose to report the analysis of the beginning and end year (2008 and 2018) while only coarsely discussing results from the entirety of the study period. In doing this, we do not capture the small scale spatial variations experienced throughout the simulation time, but mostly consider the absolute change over time. Furthermore, secondary to limitations of time and computing power, we had to prioritize internal validation runs with different sample population sizes over model calibration runs. While it is known that to develop, simulate, and run an ABM requires a substantial amount of computing power, our research significantly brings into question the role of ABM in humanitarian field applicability.

As in many complex models, inputs have a degree of uncertainty and require the significant computational methods of sensitivity analyses, model calibration, and validation, to efficiently understand real-world non-linear scenarios. Although we did not quantify uncertainty for each of the model's variable parameters due to time and resource constraints, the parameter values are not wide and thus physically constrained, and with the exceptions of the environmental indices, are mostly binary, all of which theoretically limit the degree of uncertainty in the model. Notably, four of the model's variables—slope, ethnic boundaries, artificial water sources, and land ownership—are fixed. That said, the lack of a sensitivity analysis for the remaining variable parameters—either local or global—limits the predictability of the model.

Our goal for the model was to explore how environmental, conflict, and land tenure variables can impact pastoralist livelihood and migrations. While this model is limited by a variety of factors, including data availability and resources, this research is intended to provide an exploratory window into the interactions of these domains in data-scarce environments. We also intend to inspire and inform additional thinking about next generation modelling that can be even more robust, more specific, and more precise with quantifiable uncertainties. For instance, a global sensitivity analysis, often used to quantify uncertainty in environmental

impact science, neural networks, and climate change effects, might be useful in a multi-disciplinary model such as ours to gain added confidence for targeting interventions in migratory populations.

The model also makes a number of assumptions about pastoralist behavior that while duly gleaned from informed local non-governmental service providers, have not been scientifically validated or have an evidence-based understanding of real-time migration beyond anecdotal observations. As a result, the variables are weighted based on these assumptions and migration patterns cannot be fully validated. Without clear behavioral and decision-making benchmarks, the model output is challenging to validate; rather, validation is done visually using heuristics from the social science domain literature and expert opinion. That said, the benefit of the agent-based model presented here provides some insight into pastoralist behavior and further invites theory and generates hypotheses for deeper study.

## Conclusion

The agent-based model introduced here is a useful tool to understand the behavior of individuals in a spatial and temporal context. Short of tracking individuals prospectively and manually querying their decision-making, the ABM, coupled with an ethnographic understanding of the factors and drivers of livelihood decisions, can provide a dynamic view of population ecosystems. In the case of nomadic pastoralists in the Horn of Africa where the history of seasonal migration lends itself to a simulated model, the ABM affords the ability to study the complexities of individual attributes in an aggregated and collective fashion with the added benefit of exploring layers of geographic and sociological variables.

In the humanitarian sphere, two critical independent and interdependent variables have the potential to trigger migration: climate-related environmental conditions, of which there are several, and conflict. Somaliland, with its long history of ethnic conflict over land use, its exposure to drought and water scarcity, and a basal level of livelihood migration amongst its nomadic pastoralists, offers a crucible in which to explore how these variables interplay in a model, assuming non-linearity and the need for non-parametric approaches to spatial analysis.

The use of ABM to understand pastoral migration in Somaliland, and the ways in which it changes in response to environmental variability and conflict, highlighted several important findings. This model was developed with limited data sources and capacity for large scale computational analysis, and still demonstrated the potential value added of the methodology in understanding pastoralist migration. In an attempt to evolve this model into one that is better representative of reality, the researchers intend to further refine the model, validating behavioral assumptions and including information and decisional pathways regarding economic markets, the effects of livestock disease, pastoral adaptation techniques, and resource depletion to develop a more holistic view of the pastoralist system in Somaliland and Puntland.

## Supporting information

**S1 Acronyms.**
(DOCX)

**S1 Data Sources.**
(DOCX)

**S1 File.**
(DOCX)

## Author Contributions

**Conceptualization:** Erica L. Nelson, Saira A. Khan, P. Gregg Greenough.

**Data curation:** Saira A. Khan, Swapna Thorve.

**Formal analysis:** Erica L. Nelson, Saira A. Khan, Swapna Thorve.

**Investigation:** Erica L. Nelson, Saira A. Khan.

**Methodology:** Erica L. Nelson, Saira A. Khan, Swapna Thorve, P. Gregg Greenough.

**Project administration:** Erica L. Nelson, Saira A. Khan.

**Software:** Swapna Thorve.

**Supervision:** P. Gregg Greenough.

**Validation:** Erica L. Nelson, Saira A. Khan, Swapna Thorve.

**Visualization:** Erica L. Nelson, Saira A. Khan.

**Writing – original draft:** Erica L. Nelson, Saira A. Khan.

**Writing – review & editing:** Erica L. Nelson, Saira A. Khan, P. Gregg Greenough.

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
