## [Decision Letter · Decision Letter 0]

11 Apr 2020

PONE-D-20-00682

Modeling pastoralist movement in response to environmental variables and conflict in Somaliland: Combining agent-based modeling and geospatial data

PLOS ONE

Dear Dr. Nelson,

Thank you for submitting your manuscript to PLOS ONE. After careful consideration, we feel that it has merit but does not fully meet PLOS ONE’s publication criteria as it currently stands. Therefore, we invite you to submit a revised version of the manuscript that addresses the points raised during the review process.

I concur with the recommendations of both reviewers. The revised version of the paper must clearly address methodological issues identified by both reviewers. In addition to reporting model calibration and validation results, global sensitivity analysis is highly recommended as well. Novel insights from the proposed ABM and its generalizability potential to simulate other pastoral societies must also be addressed in the discussion section.

We would appreciate receiving your revised manuscript by May 26 2020 11:59PM. To enhance the reproducibility of your results, we recommend that if applicable you deposit your laboratory protocols in protocols.io, where a protocol can be assigned its own identifier (DOI) such that it can be cited independently in the future. For instructions see: http://journals.plos.org/plosone/s/submission-guidelines#loc-laboratory-protocols

We look forward to receiving your revised manuscript.

Kind regards,

Asim Zia, Ph.D.

Academic Editor

PLOS ONE

Journal Requirements:

2. Please provide additional information about the Nairobi workshop for the delineation of private and public land. In particular, please provide additional information about participant inclusion criteria and how consensus was achieved.

4. Thank you for stating the following in the Financial Disclosure section:

"This project was funded by the National Center for Civic Innovation Inc (https://www.fcny.org/fcny/about/ncci/) acting through and on behalf of The Governance Lab (“GovLab”; https://www.thegovlab.org/). The project grant (no available grant #) funded PGG, ELN, and SAK to conduct this research. This project was originally sponsored by the World Bank (https://www.worldbank.org/).

We note that you received funding from a commercial source: 'National Center for Civic Innovation Inc'

5. We note that Figures 1, 3, 4 and 5 in your submission contain [map/satellite] images which may be copyrighted. All PLOS content is published under the Creative Commons Attribution License (CC BY 4.0), which means that the manuscript, images, and Supporting Information files will be freely available online, and any third party is permitted to access, download, copy, distribute, and use these materials in any way, even commercially, with proper attribution. For these reasons, we cannot publish previously copyrighted maps or satellite images created using proprietary data, such as Google software (Google Maps, Street View, and Earth). For more information, see our copyright guidelines: http://journals.plos.org/plosone/s/licenses-and-copyright.

a)    You may seek permission from the original copyright holder of Figures 1, 3, 4 and 5 to publish the content specifically under the CC BY 4.0 license.  

Additional Editor Comments (if provided):

I concur with the recommendations of both reviewers. The revised version of the paper must clearly address methodological issues identified by both reviewers. In addition to reporting model calibration and validation results, global sensitivity analysis is highly recommended as well. Novel insights from the proposed ABM and its generalizability potential to simulate other pastoral societies must also be addressed in the discussion section.

Reviewers' comments:

Reviewer's Responses to Questions

**Comments to the Author**

1. Is the manuscript technically sound, and do the data support the conclusions?

Reviewer #1: Partly

Reviewer #2: Partly

2. Has the statistical analysis been performed appropriately and rigorously? 

Reviewer #1: N/A

Reviewer #2: No

3. Have the authors made all data underlying the findings in their manuscript fully available?

Reviewer #1: Yes

Reviewer #2: Yes

4. Is the manuscript presented in an intelligible fashion and written in standard English?

Reviewer #1: Yes

Reviewer #2: Yes

5. Review Comments to the Author

Reviewer #1: Summary: The authors present an ABM of pastoralist mobility in Somaliland, designed to assess the impacts of environmental variability and conflict on pastoralist movement and the potential for pastoralist drop-out due to loss of herds. This is an important topic for reasons that the authors describe, and I think that the authors frame the problem appropriately and provide a clear presentation of their work. As there is a need for more published descriptions of ABMs on this topic I would like to see the paper published. That said, I have several comments on formulation and experimental design that I would like to see addressed prior to publication. It appears that the authors were severely limited by computing capacity in this study, and for this reason I'm not sure that they can address my concerns with model and simulation design. The "right" solution to that problem is for the authors to get access to more computing power; this is a big ABM, but it's not historically big, and assuming it is coded in a reasonable fashion it should be possible to do more complete simulations and analysis of output given reasonable computing resources. If this is not possible then I don't intend to block publication, for the reasons stated above, but these limitations really do handicap the value of the study.

General comments:

1. It appears that the only two time-varying drivers of migration are SAVI and conflict data. The nature of their temporal variability is not presented in the results, but it should be. This could be as simple as district-averaged SAVI and district total # of conflicts by year and season, 2008-2018. This would allow the reader to see trends and variability relevant to the authors' hypotheses, and would allow the authors to speak to these proposed drivers in a more concrete way. A time series of the average favorability score by district would also be helpful.

2. There's no way for the reader to understand the relative importance of conflict and environment based on the results as they are presented. One way to deal with this would be to repeat the simulation but with interannual environmental variability eliminated (i.e., climatology), which in this case I think would mean calculating favorability using climatologically averaged SAVI. The same could be done with a constant conflict rate. Doing this would require performing additional simulations, but since it does not require any increase in computational demand over the baseline model I would hope that the authors are able to do it.

3. I appreciate that computing limitations can be a problem, but it seems very odd to me that the authors are only able to show 2008 and 2018 output. This limitation really drives down the interpretability of the study and the richness of its results. I strongly encourage the authors, if it is at all possible, to find a computing solution that will allow them to look at output from every year in the simulation. Without this the results are really quite thin, and they can't be applied to extract an understanding of conflict and environment impacts on ABM dynamics in any robust way.

Specific comments:

line 132-133: I appreciate the need to develop models for Somaliland, but in what other ways does the research in this paper differ from the models listed above in this paragraph? The paper would be more compelling if the ABM work were framed in terms of innovation (or at least alternative formulation) relative to existing ABM of pastoralism in East Africa.

line 174: What is the source of the NDWI data? Google Earth is a platform, not a primary data source. Some information on these data is required, since the authors later assume that all mapped waterbodies are indicative of rainy season water extent in the study period (with both rainy seasons having the same water coverage(?)), and that water coverage is negligible in the dry season.

line 291-295: This is a critical rule, and I don't quite understand the logic. Why wouldn't a pastoralist choose to move to public land, even if it is not the most favorable grid cell, after negotiations fail, rather than letting their herd die and becoming an IDP?

line 343 et seq.: How does this approach account for the possibility that agents interact with each other? It would seem that simulating only 10% of households would make interactions--namely, competition for a favorable grid cell--less common, such that behaviors and resulting spatial distributions in a simulation that included 100% of households might be systematically different from a simulation that includes only 10% of households. Some explanation on this point would be useful.

line 495: "given the assumption that..." Is this an assumption or a known aspect of pastoralism in this area? I appreciate that there is little data on pastoral activities in this region, but surely there must be at least narrative accounts of seasonal migration patterns?

line 555: I don't understand why it's not possible to look at model output at times other than 2008 and 2018. This strikes me as a huge limitation, since it means the authors can't look at whether the climatic events that they hypothesize to be drivers of change actually correspond to times and locations of significant population change. I appreciate that this is a big model, but is it really impossible to store output? I would expect that an ABM on this scale could be run on a reasonably powerful server or cloud service in a way that would allow the operator to store output.

Reviewer #2: My biggest concern about the manuscript is the missing of validation. Even though empirical validation is difficult due to data paucity, it is still necessary to carry out some theoretical validations, that is to use the model to simulate some well-known and/or widely accepted theories. Without validation, the exercise is just a baseless numerical game.

Additionally, the spatial analysis techniques are quite rudimentary and do not provide much insights. The necessary sensitivity analysis for such models is also missing. It is rather disappointing that the model does not discover any thresholds, non-linearity, emergence, or other phenomena of complexity that make the ABM worthwhile.

Overall, I think the model needs to be fundamentally improved in order to produce scientifically valuable results.

6. PLOS authors have the option to publish the peer review history of their article (what does this mean?). If published, this will include your full peer review and any attached files.

Reviewer #1: No

Reviewer #2: No

---

## [Author Response · Author response to Decision Letter 0]

5 Oct 2020

Dear Editors and Reviewers,

We thank you for your thorough review and opportunity to revise and re-submit our manuscript entitled, “Modeling pastoralist movement in response to environmental variables and conflict in Somaliland: Combining agent-based modeling and geospatial data”. We have revised the manuscript based on your reviews and recommendations, which we believe has substantially enhanced the overall manuscript. Our responses to the editor and reviewer’s comments are included in this revision as a separate document for formatting reasons. 

Thank you for considering this resubmission. 

Warm regards,

Erica Nelson

---

## [Decision Letter · Decision Letter 1]

7 Dec 2020

Modeling pastoralist movement in response to environmental variables and conflict in Somaliland: Combining agent-based modeling and geospatial data

PONE-D-20-00682R1

Dear Dr. Nelson,

We’re pleased to inform you that your manuscript has been judged scientifically suitable for publication and will be formally accepted for publication once it meets all outstanding technical requirements.

Kind regards,

Asim Zia, Ph.D.

Academic Editor

PLOS ONE

Additional Editor Comments (optional):

Reviewers' comments:

Reviewer's Responses to Questions

**Comments to the Author**

1. If the authors have adequately addressed your comments raised in a previous round of review and you feel that this manuscript is now acceptable for publication, you may indicate that here to bypass the “Comments to the Author” section, enter your conflict of interest statement in the “Confidential to Editor” section, and submit your "Accept" recommendation.

Reviewer #1: All comments have been addressed

2. Is the manuscript technically sound, and do the data support the conclusions?

Reviewer #1: Yes

3. Has the statistical analysis been performed appropriately and rigorously? 

Reviewer #1: N/A

4. Have the authors made all data underlying the findings in their manuscript fully available?

Reviewer #1: Yes

5. Is the manuscript presented in an intelligible fashion and written in standard English?

Reviewer #1: Yes

6. Review Comments to the Author

Reviewer #1: (No Response)

7. PLOS authors have the option to publish the peer review history of their article (what does this mean?). If published, this will include your full peer review and any attached files.

Reviewer #1: No

---

## [Editor Report · Acceptance letter]

11 Dec 2020

PONE-D-20-00682R1 

Modeling pastoralist movement in response to environmental variables and conflict in Somaliland: Combining agent-based modeling and geospatial data 

Dear Dr. Nelson:

I'm pleased to inform you that your manuscript has been deemed suitable for publication in PLOS ONE. Congratulations! Your manuscript is now with our production department. 

Kind regards, 

on behalf of

Professor Asim Zia 

Academic Editor

PLOS ONE